# Association between quantitative sensory testing and pain or disability in paediatric chronic pain: protocol for a systematic review and meta-analysis

Daniel Eric Schoth,[1] Markus Blankenburg,[2] Julia Wager,[3] Philippa Broadbent,[1] Jin Zhang,[1] Boris Zernikow,[3] Christina Liossi[1,4]

[1]School of Psychology, University of Southampton, Southampton, UK
[2]Faculty of Health, University of Witten/Herdecke, Witten, Germany
[3]Universitat Witten/Herdecke, Witten, Germany
[4]Paediatric Psychology, Great Ormond Street Hospital for Children NHS Foundation Trust, London, UK

**Correspondence to**
Professor Christina Liossi;
C.Liossi@soton.ac.uk

## ABSTRACT

**Introduction** This protocol describes the objective and methods of a systematic review of the association between quantitative sensory testing (QST) measures and pain intensity or disability in paediatric chronic pain (PCP). The review will also assess whether the relationship strength is moderated by variables related to the QST method and pain condition; the use of QST in PCP (modalities, outcome measures and anatomical test sites as well as differentiating between pain mechanisms (eg, neuropathic vs nociceptive) and in selecting analgesics); the reliability of QST across the paediatric age range; the ability of QST to differentiate patients with chronic pain from healthy controls; and differences between anatomical test sites.

**Methods and analysis** Medline, PsycINFO, CINHAL, Web of Science, Scopus, Cochrane Library and OpenGrey will be searched. English language studies will be eligible if they recruit a sample aged 6–24 (inclusive) with chronic pain, including primary and secondary pain; apply at least one of the following QST modalities: chemical, electrical, mechanical (subgroups include pressure, punctate/brush and vibratory) or thermal stimulus to measure perception of noxious or innocuous stimuli applied to skin, muscle or joint; use a testing protocol to control for stimulus properties: modality, anatomical site, intensity, duration and sequence. Following title and abstract screening, the full texts of relevant records will be independently assessed by two reviewers. For eligible studies, one reviewer will extract study characteristics and data, and another will check for accuracy. Both will undertake independent quality assessments using the Appraisal Tool for Cross-Sectional Studies. A qualitative synthesis will be presented with discussion centred around different QST modalities. Where eligible data permit, meta-analyses will be performed separately for different QST modalities using comprehensive meta-analysis.

**Ethics and dissemination** Review findings will be reported in a peer-reviewed journal and presented at conferences. The study raises no ethical issues.

**PROSPERO registration number** CRD42019134069.

## INTRODUCTION

Paediatric chronic pain (PCP) is defined as persistent or recurring pain of any aetiology

### Strengths and limitations of this study

► This systematic review can give us a greater understanding of the association between quantitative sensory testing (QST) (in accordance with either the DFNS (German Research Network on Neuropathic Pain) protocol or other protocols) and pain intensity or disability in paediatric chronic pain and could inform the use of QST in clinical practice.
► This systematic review will follow robust guidelines and the quality of included articles will be assessed using validated tools.
► The heterogeneity of the included studies, which may use different quantitative sensory measures across different chronic pain conditions and age range, could limit the overall data synthesis.

lasting for longer than 3 months.[1 2] A cross-sectional study of 561 schoolchildren aged 8–16 (mean age 11.89) showed a PCP prevalence of 37.3%, of which 29.3% reported pain in two or more sites.[1] Prevalence rates do vary across reports and conditions however, as demonstrated in a systematic review of 41 articles (eg, 4%–40% for musculoskeletal pain, 8%–83% for headache and 4%–53% for abdominal pain) which may be due, in part, to inconsistent definitions of chronic pain adopted by individual studies.[3] An analysis of the 2016 National Survey of Children's Health in the USA estimated a population prevalence of PCP to be 6% based on a sample of 43 712 children aged 0–17 years.[4]

Despite variations in precise prevalence estimates, PCP is a significant clinical problem worldwide[5] and can have enormous impacts on the child's quality of life.[6] Compared with their healthy peers, children with chronic pain report substantially worse quality of life in physical, emotional, social and school functioning[7] and frequently experience disordered sleep.[8] A purely biomedical model of

PCP is outdated, and instead the severity, impact and experience of chronic pain are now recognised to be shaped by a complex interaction of biological, psychological and social factors.[9] Multimodal interdisciplinary interventions are therefore essential in the treatment of severe PCP.[10 11] A recent review of 21 studies highlights the effectiveness of interdisciplinary interventions in PCP management. Patients receiving interdisciplinary interventions reported significant improvements preintervention to postintervention in pain intensity, functional disability, anxiety, depression, catastrophising, school attendance, school functioning and pain acceptance, and significantly lower pain intensity 0–1 month postintervention compared with patients randomised to control groups.[12]

Quantitative sensory testing (QST) refers to a range of non-invasive techniques for exploring somatosensory processing, including positive and negative sensory phenomena.[13 14] QST is psychophysical in nature, relying on the individual's subjective report to a standardised stimulus.[13 15] QST might provide unique information about the functional status of the somatosensory system and therefore better guide pain treatment. Although QST is commonly used in adult populations, QST is also considered feasible, reliable and reproducible in children, including children as young as 4 years old when using thermal stimuli.[16] In recent years a small but growing number of published studies have explored the use of QST in PCP populations, although calls have been made to expand this further.[17] Similar to other psychophysical methods and to enable the comparison with published normative values and between different QST labs as well as follow-up examinations, QST requires the active participation of the young patient, a standardised assessment using standardised instructions and calibrated stimuli applied by highly trained investigators. A number of protocols are followed to this end with the protocol proposed by the German Research Network on Neuropathic Pain (DFNS) the most well developed. DFNS includes examination of warm and cold detection and pain thresholds including the assessment of paradoxical heat sensation during alternating temperature stimuli (using a thermostat), mechanical detection thresholds for light touch (using a set of von Frey filaments) and for vibration (using a Rydel-Seifter tuning fork), mechanical pain thresholds for pinprick (using pinprick stimulators of different intensities) and for blunt pressure (using a pressure algometer) as well as a stimulus-response function for mechanical pain sensitivity to pinprick stimuli, dynamic mechanical allodynia for the stimuli of different intensities (using a cotton wool, a Q tip and a standardised brush) and temporal summation for pinprick stimuli.[18]

The primary objective of this systematic review and meta-analysis is to examine the relationship between established QST measures and pain intensity or disability in PCP. The second objective is to assess whether the strength of the relationship is moderated by variables related to the QST method and pain condition. Furthermore, this systematic review and meta-analysis will examine: (1) the use of QST in PCP (modalities, outcome measures and anatomical test sites, differentiating between pain mechanisms (eg, neuropathic vs nociceptive) and in selecting analgesics); (2) the reliability of QST across the paediatric age range; (3) the ability of QST to differentiate patients with chronic pain from healthy controls; and (4) differences between anatomical test sites.

## METHODS

This protocol was developed in-line with Preferred Reporting Items for Systematic Review and Meta-Analysis Protocolsguidelines.[19 20] Any amendments to the protocol will be recorded on PROSPERO.

### Literature search

We will search the following electronic bibliographic databases: Medline, PsycINFO, CINHAL (title, abstract), Web of Science (title), Scopus (article title, abstract and keywords) and Cochrane Library (title, abstract and keywords). A search of non-traditional publications, commonly known as 'grey literature', will be conducted using Open Grey. All databases will be searched from database inception to the date of search. Only studies published in the English language will be included. The reference lists of all eligible papers will be hand-searched for relevant studies. The search strategy will include three concept blocks pertaining to the type of pain (*pain\*, ache\*, abdominal pain, arthritis, headache\*, musculoskeletal pain, fibromyalgia*), study population (*boy\*, girl\*, child\*, teenager\*, youth\*, adolescen\*, young\*, schoolchild\*, school child\*, juvenil\*, paediatric, pediatric*) and QST (*quantitative sensory test\*, sensory test\*, QST, threshold, tolerance, pressure, electrical, warm, heat, hot, cool, cold, mechanical, hyperalgesia, hyperaesthesia, allodynia, sensitisation, sensitivity*) (note: the asterisk is used as a truncation command to search the root word along with any endings, for example *child\** would retrieve both *child* and *children*).

### Inclusion criteria

Studies will be eligible for inclusion if they meet the following criteria:
1. Available in the English language.
2. Recruits a sample aged 6–24 (inclusive) with chronic pain, including primary pain (ie, pain in one or more anatomic regions that has lasted for longer than 3 months and cannot be better explained by any other chronic pain condition) and secondary pain (ie, pain that has lasted longer than 3 months that is symptomatic of another health condition).[21] In cases where the sample age range includes individuals older than 24 years of age, the record will only be eligible for inclusion if separate data are presented for the age group of interest to the present review. While the adolescent age range typically extends to 18 years, the transition period from childhood to adulthood is now argued to occupy a greater portion of the life course than before,

and therefore an upper age limit of 24 years corresponds more closely to adolescent growth and popular understandings of this life phase.[22]

3. Applies at least one of the following QST modalities: chemical, electrical, mechanical (subgroups will include pressure, punctate/brush, and vibratory) or thermal stimulus, in order to measure perception of noxious or innocuous stimuli applied to skin, muscle or joint.

4. Uses a QST testing protocol that allows reliable replication of the procedure followed and which controls for stimulus properties: modality, anatomical site, intensity, duration and sequence.

Study exclusion criteria for this review are:

1. Use of an invasive form of QST only (eg, rectal barostat).

2. Examination of pain modulation only.

3. Providing correlations between pain intensity/disability and QST composite scores (ie, local and remote site combined) only.

QST is defined as a method that quantifies the magnitude of physical stimuli (eg, pressure, heat, cold, vibration, electrical current) that is required to determine a specific perception (ie, threshold, tolerance, temporal summation, magnitude rating, wind-up, limen). For inclusion in this systematic review, the application of the physical stimulus will have to be standardised and the physical stimulus will have to be expressed in quantitative terms (eg, pressure: $kg/cm^2$; heat/cold: °C). The evoked sensory and pain perception will have to be reported subjectively by the participant (eg, yes/no for pin prick threshold) and quantitatively (eg, pressure: $kg/cm^2$; heat/cold: °C; intensity ratings using VAS, NRS or other validated scales). Correlation coefficients (Pearson's r or Spearman's q) for the relationship between QST measured locally and/or at a remote site and pain intensity/disability will have to be reported or could be calculated from the data reported in the study or from data obtained from the study authors.

## Data extraction

The titles and abstracts of retrieved records will be transferred into the digital reference manager Endnote X8. Titles will be scanned for duplication using an automated search engine within Endnote, and any duplicates will be removed. Records will then be screened by two review team members independently to identify studies that potentially meet the inclusion criteria (two independent reviewers were decided based on Preferred Reporting Items for Systematic Reviews and Meta-Analyses guidance[23] and the helpful comments from an independent reviewer of this protocol manuscript). The full texts of potentially relevant records will be retrieved, which will be independently assessed for eligibility by the same two review team members. Any disagreement between reviewers over study eligibility will be resolved through discussion with a third review author. Data will be extracted using a prepiloted template. Extracted information will include

the country in which the research was conducted; demographics and medical characteristics of PCP patients; demographics of healthy controls (where applicable); QST protocol details; the training of the QST administrators; the outcome measures recorded; and QST and other outcome results. For eligible studies, one reviewer will perform extraction of study characteristics and data, and another will check for accuracy (with a third review author where necessary). If information is missing or unclear, the corresponding author for the study in question will be contacted.

## Study quality assessment

Two review authors will independently assess study quality using AXIS.[24] AXIS features 20 questions which are answered as either 'yes', 'no' or 'don't know', along with spaces for comments on each judgement provided. Although similar to the STROBE statement (a 22-item checking for use with observational (cohort, cross-sectional, case–control) studies,[25] AXIS provides an assessment of both study design and risk of bias. Inter-rater reliability between the two reviewers will be calculated using Cohen's kappa, where a kappa value of 0.80 or above is considered a strong level of agreement and indicates high inter-rater reliability.[26] Any disagreements between the review authors will be resolved by discussion, with involvement of a third review author where necessary.

## Statistical analysis

Meta-analysis will be performed to address the study objectives where data from two or more eligible studies is available.[27] Data will be grouped into clinically meaningful diagnostic categories (eg, abdominal pain, migraine, tension-type headache, fibromyalgia, arthritis) and analysed separately. Due to developmental and sex differences in somatosensory perception[28 29] data will be grouped and analysed separately where possible for males and females, and by appropriate age groups. Regarding the latter we anticipate we will analyse two groups (1) 6–20 and (2) 21 and over, unless there is enough data and we are able to follow and modify the groupings suggested by Blankenburg and colleagues[28] (ie, younger children (6–8 years), older children (9–12 years), younger adolescents (13–16 years) and older adolescents (17–24 years)). Where data are not provided separately per age group (eg, if the sample age ranges from 18 to 24), we will contact the study authors and request separate data. If separate data are not provided or available we will not be able to include the data in any relevant meta-analyses. Data will also be grouped into local (ie, the site of pain) and non-local (ie, a non-painful site) categories and will initially be analysed separately, although based on the suggestion of an independent reviewer will also perform an analysis of all sites combined where possible. Analyses will be conducted separately by type of stimuli (ie, pressure, mechanical, heat, cold, vibration, electrical, chemical) due to different pathways mediating the nociceptive and pain experience.

All statistical analyses will be performed using comprehensive meta-analysis (CMA V.3.0[30] and Power and Sample Size Calculations V.3.0.43 software. CMA converts correlation coefficients to Fisher's z scores, with all analyses performed using these transformed values.[31] Similar to a recent review of QST in spinal pain,[32] Pearson's r or Spearman's rank coefficient will be synthesised together. Unlike Pearson's product-moment correlation coefficient, which requires the assumption that the relationship between the variables is linear and variables to be measured on interval scales, Spearman's rank correlation coefficient is a non-parametric (distribution-free) rank statistic. It assesses how well an arbitrary monotonic function can describe a relationship between two variables, without making any assumptions about the frequency distribution of the variables and can be used for variables measured at the ordinal level. We will use 0.25, 0.5 and 0.75 as cut-off points to interpret the strength of the relationship as little or zero (0.00–0.25), fair (0.26–0.50), moderate to good (0.51–0.75) and good to excellent (above 0.75).[33]

Heterogeneity will be examined using Cochrane's Q test and the $I^2$ statistic. With Cochrane's Q, a significant result is indicative of heterogeneity. The $I^2$ statistic describes the percentage of variability in effect estimates due to heterogeneity as opposed to sampling error.[34] The Cochrane Handbook provides the following rough guide for interpretation of the $I^2$ statistic: 0%–40%: might not be important; 30%–60%: may represent moderate heterogeneity; 50%–90%: may represent substantial heterogeneity; 75%–100%: considerable heterogeneity.[34] We will conduct sensitivity analyses where appropriate to verify the robustness of the study conclusions, and for more than 10 studies, we will use funnel plots to detect potential reporting biases and small-study effects. Given sufficient availability of data, mixed-effects metaregression will be used to formally assess the effect of the potential categorical predictors (QST testing site, pain condition, pain type, type of pain induction stimulus) on the strength of the relationship between pain threshold and pain intensity/disability. The effect of each categorical predictor will be tested in a separate metaregression.

The ability of QST to detect somatosensory differences between young people with chronic pain and healthy controls will be assessed using standardised mean differences (SMDs). The SMD will be calculated as the mean difference between groups divided by the SD of the specific QST measure for all participants pooled across both groups. The SMD expresses the difference between groups in QST measures as multiples of the observed SD. The SMD is a ratio, with numerator and denominator in the same units as the original measurement, therefore has no units, does not depend on the original measurement scale, and allows a direct comparison of the effect across studies that used different QST measures. By convention, an SMD or d=0.2 is considered to be small, d=0.5 moderate and d=0.8 large in size.[35] Each study will be screened for intraclass correlation coefficients (ICCs)

describing the test–retest reliability of QST. The reliability of the measurement across test occasions will be rated excellent if the ICC >0.75, adequate if 0.40–0.74, and poor if <0.40.[36]

## Patient and public involvement

Consultation with young people, parents and healthcare professionals has been used to determine their perception of the clinical utility of QST in chronic pain. It is based on their perspectives that this systematic review was deemed timely and crucial to conduct to inform further research work and implementation of QST in routine clinical practice.

## DISCUSSION

This systematic review will be the first to investigate the association between QST and pain intensity or disability in PCP. The review findings will be used to inform our ongoing work to develop a core outcome set for specialist PCP services across the UK.

## ETHICS AND DISSEMINATION

As this is a systematic review of published literature, ethical approval will not be sought. We will publish the protocol and our findings in peer-reviewed journals aimed at PCP clinicians and researchers as well as health commissioners. We will present our work at national and international meetings focused on PCP.

**Acknowledgements** We would like to thank Holly Brown, Navin Down, Victoria Gibbons and Olivia Ranger (BSc Psychology students studying at the University of Southampton who undertook their 3rd year dissertation projects at the Pain Research Laboratory) for performing a scoping review of relevant literature that informed the development of this protocol.

**Contributors** CL conceived the idea, planned and designed the study protocol. CL and DES planned the data extraction and statistical analysis and wrote the first draft; MB, JW, BZ, JZ and PB provided critical insights. All authors have approved and contributed to the final written manuscript.

**Funding** This development of this protocol was supported by ESRC (Grant number: RES-000-22-4128). This funding source had no role in the design of this review and will not have any role during its execution, analyses, interpretation of the data or decision to submit results.

**Competing interests** None declared.

**Patient consent for publication** Not required.

**Provenance and peer review** Not commissioned; externally peer reviewed.

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
