## [Reviewer comments · BMJ Open]

ARTICLE DETAILS

TITLE (PROVISIONAL)	The association between quantitative sensory testing and pain or disability in paediatric and young adult chronic pain: Protocol for a systematic review and meta-analysis
AUTHORS	Schoth, Daniel; Blankenburg, Markus; Wager, Julia; Broadbent, Philippa; Zhang, Jin; Zernikow, Boris; Lioffi, Christina

VERSION 1 - REVIEW

REVIEWER	David Moore Liverpool John Moores University UK
REVIEW RETURNED	18-Jun-2019

GENERAL COMMENTS	The purpose of this review will be to understand psychophysical pain responses of children with chronic pain compared to controls. This is a useful and timely review which, on completion, has the potential to greatly help understanding of this population. I would like to ask the authors to consider a few issues however at this time which may be important to consider in commission of this research. 1) The population of interest here runs from 6-24, I wonder if this is the most appropriate boundaries to use, I acknowledge that adolescents is now considered to last until a later period in development however a number of the published normative studies have sought to start studies from 21 years of age, this might therefore result in a bias towards this more commonly studied age group. I wonder if more impact/novelty will be obtained with a focus of the younger population. 2) I would be interested to read more about the selection of relevant tasks for review. The introduction focusses heavily on discussion of the DNFS protocol however the methods seem to be suggesting that any psychophysical approach is appropriate. Can the authors clarify how the methodological decisions have been arrived at. 3) The reporting of the decision to conduct a meta-analysis appears a little out of the blue. If this is a plan I feel that a greater clarity about how groups/approaches are going to be organised. i.e. it may not be appropriate to combine thermal and mechanical pain modes here at these are mediated by very different neural pathways, further different psychophysical approaches may result in greatly varying outcomes. Can the authors comment further here.
--

	4) Further will this review be designed to simply compare those in pain and those not, or will different conditions be separated. I would suggest that at the least nociceptive and neuropathic pains should be considered separately here as these will likely result in very different presentations of thresholds.
--	---

REVIEWER	Hadas Nahman-Averbuch Cincinnati Children's Hospital, Cincinnati, Ohio, USA
REVIEW RETURNED	25-Jun-2019

GENERAL COMMENTS	This paper describes the methods that will be used to conduct a systemic review and, if possible, a meta-analysis, on the relationships between QST measures and pain intensity and disability in chronic pain pediatric patients. I have a few comments:  1. How many eligible studies will be required to perform a meta-analysis? 2. In the literature search, the authors use the sign *. Please provide an explanation of this sign for readers who may not be familiar with it. 3. Inclusion criteria number 4 is not clear. What kind of testing protocol is needed? Is having information about modality, anatomical site etc. enough for studies to be included in the review and meta analysis? 4. The authors do not mention an inclusion criterion for testing the relationship between QST and pain intensity/disability, which is the primary aim of the study. 5. Studies that have combined QST scores for local and remote sites will be excluded. This should appear in the list of exclusion criteria. Also, depending on the number of studies, is it possible to conduct an analysis of all sites (i.e combined) and separate analyses based on local and remote areas? 6. The authors chose to include subjects between the age of 6-24. I do not agree with the authors' rationale to include subjects older than 18. However, if the authors chose to include these subjects they should change the title and clearly mention that this is a review in pediatric and young adults. A separate meta-analysis can be conducted including young adults, but I would keep the focus on pediatric patients. 7. Will studies that examined pain modulation using QST be included? For example, conditioned pain modulation? 8. Based on the PRISMA statement, it is preferred to include more than one investigator screening all records (not just 10% of records). 9. The authors mention separate analyses based on age group. Please specify how many groups and the age ranges for each group.
---

VERSION 1 – AUTHOR RESPONSE

Response to Reviewers

We would like to thank Drs Moore and Nahman-Averbuch for their time reading our manuscript and providing constructive recommendations. Below follows our point by point response.

Reviewer: 1 David Moore

The purpose of this review will be to understand psychophysical pain responses of children with chronic pain compared to controls. This is a useful and timely review which, on completion, has the potential to greatly help understanding of this population. I would like to ask the authors to consider a few issues however at this time which may be important to consider in commission of this research.

1) The population of interest here runs from 6-24, I wonder if this is the most appropriate boundaries to use, I acknowledge that adolescents is now considered to last until a later period in development however a number of the published normative studies have sought to start studies from 21 years of age, this might therefore result in a bias towards this more commonly studied age group. I wonder if more impact/novelty will be obtained with a focus of the younger population.

Thank you. We appreciate your suggestions. We anticipate that we will analyse two groups i.e., 6-20 and 21 and over, unless there is enough data and we are able to follow and modify the groupings suggested by Blankenburg and colleagues¹ i.e., younger children (6–8 years), older children (9–12 years) and younger adolescents (13–16 years; n = 64) and older adolescents (17-24 years).

2) I would be interested to read more about the selection of relevant tasks for review. The introduction focusses heavily on discussion of the DNFS protocol however, the methods seem to be suggesting that any psychophysical approach is appropriate. Can the authors clarify how the methodological decisions have been arrived at.

Any QST testing protocol that allows reliable replication of the procedure that was followed can be included in the systematic review. The DNFS protocol is the most detailed one currently therefore we discuss it more extensively.

3) The reporting of the decision to conduct a meta-analysis appears a little out of the blue. If this is a plan I feel that a greater clarity about how groups/approaches are going to be organised. i.e. it may not be appropriate to combine thermal and mechanical pain modes here at these are mediated by very different neural pathways, further different psychophysical approaches may result in greatly varying outcomes. Can the authors comment further here.

We agree with you. We now clarify in the manuscript that we will analyse separately by type of physical stimuli (e.g., pressure, heat, cold, vibration, electrical current)

4) Further will this review be designed to simply compare those in pain and those not, or will different conditions be separated. I would suggest that at the least nociceptive and neuropathic pains should be considered separately here as these will likely result in very different presentations of thresholds.

Thank you. We will separate into clinically meaningful diagnostic categories (e.g., abdominal pain, migraine, fibromyalgia), which we now clarify in the manuscript.

Reviewer: Hadas Nahman-Averbuch

This paper describes the methods that will be used to conduct a systemic review and, if possible, a meta-analysis, on the relationships between QST measures and pain intensity and disability in chronic pain pediatric patients. I have a few comments:

1. How many eligible studies will be required to perform a meta-analysis?

Meta-analysis can be performed with as few as two studies 2, which we now mention in the manuscript.

2. In the literature search, the authors use the sign *. Please provide an explanation of this sign for readers who may not be familiar with it.

We have now provided an explanation for the use of this truncation command.

3. Inclusion criteria number 4 is not clear. What kind of testing protocol is needed? Is having information about modality, anatomical site etc. enough for studies to be included in the review and meta-analysis?

Any QST testing protocol that allows reliable replication of the procedure that was followed can be included in the systematic review. We now clarify this in the manuscript.

4. The authors do not mention an inclusion criterion for testing the relationship between QST and pain intensity/disability, which is the primary aim of the study.

The relationship between QST and pain intensity/disability is indeed our primary research question but is not necessary for a study to be included to have tested that relationship.

5. Studies that have combined QST scores for local and remote sites will be excluded. This should appear in the list of exclusion criteria. Also, depending on the number of studies, is it possible to conduct an analysis of all sites (i.e combined) and separate analyses based on local and remote areas?

We have now added in the exclusion criteria that studies that have combined QST scores for local and remote sites will be excluded. As you suggest we will, depending on the number of studies, conduct an analysis of all sites (i.e combined) and separate analyses based on local and remote areas.

6. The authors chose to include subjects between the age of 6-24. I do not agree with the authors' rationale to include subjects older than 18. However, if the authors chose to include these subjects they should change the title and clearly mention that this is a review in pediatric and young adults. A separate meta-analysis can be conducted including young adults, but I would keep the focus on pediatric patients.

Thank you. We appreciate your suggestions. We anticipate that we will analyse two groups i.e., 6-20 and 21 and over, unless there is enough data and we are able to follow and modify the groupings

suggested by Blankenburg and colleagues¹ i.e., younger children (6–8 years), older children (9–12 years) and younger adolescents (13–16 years; n = 64) and older adolescents (17-24 years).

7. Will studies that examined pain modulation using QST be included? For example, conditioned pain modulation?

Studies that examined pain modulation using QST are excluded from this review. We clarify this in the manuscript.

8. Based on the PRISMA statement, it is preferred to include more than one investigator screening all records (not just 10% of records).

Thank you. We take your point and have now decided that two investigators will screen all records.

9. The authors mention separate analyses based on age group. Please specify how many groups and the age ranges for each group.

We anticipate that we will analyse two groups i.e., 6-20 and 21 and over, unless there is enough data and we are able to follow and modify the groupings suggested by Blankenburg and colleagues¹ i.e., younger children (6–8 years), older children (9–12 years) and younger adolescents (13–16 years; n = 64) and older adolescents (17-24 years).

References

1. Blankenburg M, Boekens H, Hechler T, et al. Reference values for quantitative sensory testing in children and adolescents: Developmental and gender differences of somatosensory perception. *Pain* 2010;149(1):76-88 doi: <http://dx.doi.org/10.1016/j.pain.2010.01.011>.
2. Higgins JP, Green S. *Cochrane handbook for systematic reviews of interventions*: John Wiley & Sons, 2011.

VERSION 2 – REVIEW

REVIEWER	David Moore Liverpool John Moores University United Kingdom
REVIEW RETURNED	08-Aug-2019

GENERAL COMMENTS	I would like to start by thanking the authors for their attention to the manuscript and their responses. I feel that the manuscript is much clearer at this time. I have one small additional query which has developed as a result of the replies. The authors state that they intend to split their analysis into 21 and old and younger than 21, and potentially further dependent on sample size. I wonder if they might clarify how they would assign a study which tested, for example, participants aged 18-24? Will they try to get original data or use mean age or use some other approach?
---

VERSION 2 – AUTHOR RESPONSE

Reviewer 1: David Moore

I would like to start by thanking the authors for their attention to the manuscript and their responses. I feel that the manuscript is much clearer at this time.

Thank you very much

The authors state that they intend to split their analysis into 21 and old and younger than 21, and potentially further dependent on sample size. I wonder if they might clarify how they would assign a study which tested, for example, participants aged 18-24? Will they try to get original data or use mean age or use some other approach?

Thank you. We now clarify in the manuscript that if data are not provided separately per age group (for example if the sample age ranges from 18 – 24), we will contact the study authors and request this. If separate data are not provided or are not available we will not be able to include the data in any relevant meta-analyses.

VERSION 3 – REVIEW

REVIEWER	David Moore Liverpool John Moores University, UK
REVIEW RETURNED	04-Sep-2019

GENERAL COMMENTS	I would like to thank the authors for their attention to this manuscript and wish them all the fortune in conducting this review.
---